# Thermographic of the Microcirculation in Healthy Children Aged 3–10 Months as an Objective and Noninvasive Method of Assessment

**DOI:** 10.3390/ijerph192316072

**Published:** 2022-12-01

**Authors:** Agnieszka Ptak, Agnieszka Dębiec-Bąk, Małgorzata Stefańska

**Affiliations:** Faculty of Physiotherapy, University of Health and Sport Sciences in Wrocław, al. Ignacego Jana Paderewskiego 35, 51-612 Wrocław, Poland

**Keywords:** thermovision, microcirculation development, function

## Abstract

Background: The aim of this study was to assess if thermography as an objective and non-invasive research tool is capable of identifying the changes in the surface temperature of the body as a response to muscle stimulation in Vojta therapy. The research group consisted of children aged 3–10 months with slight abnormalities of the motor pattern, subjected to individually selected stimulation elements according to Vojta. Methods: The Vojta method of spontaneous motor assessment and the thermovision method of assessing the microcirculation properties of muscles were used for the evaluation. Results: In the study group, changes in the microcirculation parameters of the extensor muscles of the back occurred immediately after the therapy at the first examination. Conclusions: The analysis featuring an objective assessment allows physiotherapists to diagnose local temperature changes based on the effect of microcirculation parameters in the musculofascial structures. Trial registration: The research was conducted as a pilot study for a scientific project approved by the Commission for Scientific Research of the University of Health and Sport Sciences in Wroclaw No 24/2021. The study is currently in the registration process with the Australian New Zealand Clinical Trials Registry.

## 1. Introduction

Healthy newborns and infants develop as described in developmental norms. The intervals and the order in which new skills appear in the first year of life have been described by the authors as milestones [1]. Minor spontaneous motor abnormalities can be observed and classified as “no expected normal physical development, unspecified” (code R62.9) by the *International Classification of Diseases 10* (*ICD 10*). The observed abnormalities in the assessment present as a lack of support and extension function of the upper limbs (deficit of motor functions adequate to age) with accompanying problems in food intake and disorders of the digestive system. Such a diagnosis results in referring the patient to a physiotherapeutic process [2,3,4]. When the support and extension function of the upper limbs is limited, the postural muscles, which play a key role in building the stabilization of the whole body as a starting point for subsequent developmental milestones, are disrupted One of the methods allowing for a qualitative assessment and therapy of spontaneous motor skills is the Vojta method [5,6]. According to Vojta, the main principle of reflex locomotion is the activation of postural muscles through isometric contraction during stimulation. Constant stimulation affects muscles, joints, tendons and ligaments. Moreover, the Vojta reflex locomotion is associated with exteroceptors and enteroceptors and becomes a source of afferent stimulation that passes into the central nervous system [5,7]. The Vojta method activates core and deep muscles, regulating postural stabilization and increases the strength of spine rotation, thus improving the ability to control posture [7]. The objective assessment of muscle activity in pediatric patients in the early stage of development is difficult due to the lack of logical contact with the patient and lack of objective assessment tool. The changes taking place in the distribution of the body surface temperature as a result of muscle activity after stimulation with the Vojta method may be illustrated by a thermographic examination [8,9].

The energy required by the muscles to do the work comes from the chemical reactions that take place in the cells during contraction. More energy is needed for contraction than for muscle relaxation. Heat production is caused by metabolism, which changes under the influence of the activity and work of the muscle. Heat is produced in various tissues and is removed by the circulatory system and the blood that carries it.

The skin flow plays an essential role in the heat-release process. As the temperature rises, there is increased skin flow, and heat transfer through radiation, conduction, convection and evaporation is accelerated. This may have an impact on the ability to perform exercise or muscle activity. Children and the elderly react differently to temperature changes, and their thermoregulatory system compensates for the increase or decrease in temperature in the external environment worst, which can be observed in the infrared emissivity in the thermal imaging method [10,11,12,13,14].

Therefore, the use of thermal imaging recording may be useful for the analysis of the efficiency of thermoregulatory processes in the human body.

The method of recording the emission of infrared radiation enables the non-contact performance of the test. Thermal imaging recording is safe, non-invasive for the examined person and repeatable. It has found its application in biomedicine as a method of supplementary diagnostics. Thermal imaging can guide initial diagnostics, monitor pathophysiological processes and control therapy. Thermal imaging is also used in many medical specialties, including laryngology, oncology, ophthalmology, dentistry, angiology and surgery, as well as in dermatology to assess the extent of burns and wound healing [12,15,16].

The aim of this study was to assess if thermography as an objective and non-invasive research tool is capable of identifying the changes in the surface temperature of the body as the response to muscle stimulation in Vojta therapy. 

## 2. Materials and Methods

### 2.1. Research Group

The study group consisted of 22 healthy, full-term infants (8 girls and 14 boys) with an average age of 7 months (±3.29 months), rated at birth at an average of 9 points on the APGAR scale (±1 point). The children were referred by a pediatrician for a physiotherapeutic consultation due to minor abnormalities in spontaneous motor movement, defined as muscle tension disorders, observed during patronage visits (4–6 weeks of age). The time of starting the therapy depended on the waiting time for a therapeutic visit. Currently, the estimated waiting time for starting therapy is 6–8 months. The level of motor development in infants remained at a similar level due to the motor deficits presented from the beginning. In all infants, in the assessment of spontaneous motor movement, a reclining (straightening) position of the head was observed, as a result of which the infants showed no or age-inadequate support extension function of the upper limbs.

The inclusion criteria included an Apgar score of 8–10, good general health and the inherence of a minor qualitative disturbance of spontaneous motor skills diagnosed by a pediatrician. The exclusion criteria included prematurity and postural asymmetry as well as neurological and genetic deficits and disorders.

The parents of all included children were informed of the purpose of the study and manner of execution. Written consent to their children’s participation in the study and to the anonymous publication of results has been given. The project was approved by the Commission for Scientific Research of the University of Health and Sport Sciences in Wroclaw No 24/2021.

### 2.2. Research Methods

The children underwent a physiotherapeutic evaluation at the first visit according to Munich Functional Development Diagnostics. Due to the occurrence of minor spontaneous motor disorders in all examined patients, consisting in the lack of or inadequate-to-developmental-age support extension function, they underwent therapy consisting of stimulation in accordance with the methodology of the Vojta method. The exercise was breast-zone stimulation. The breast zone is located between either the 5th and 6th ribs or the 6th and 7th ribs. The starting position for breast-zone stimulation was supine with the head turned 30° in the direction of the stimulation, with the extremities lying naturally on the floor. During the first session, the therapist shows the stimulation to the parents, and then the parents repeat it. The stimulation takes place twice on both sides for 30 s a side. The stimulation was conducted by a therapist certified by Internationale Vojta Geselshaft. 

The imaging examination was performed once before and after the first treatment session. Thermograms were taken directly after therapy.

The ThermoVision FLIR SYSTEM T335 thermal imaging camera and the Therma CAM Researcher Pro 2.10 computer software for the analysis of thermal images were used to record the surface body temperature. The thermal imaging camera (FLIRT335) with a resolution of 320 × 240 pixels, used in the presented research project, has a built-in auto-calibration system, ensuring the appropriate homogeneity of the image characteristics. The system is activated 20–30 min before each session in order to prepare the camera for thermal imaging so that the electronics do not cause fluctuations or changes in the recorded image.

The software settings are the same and each measurement session is validated before the target recording is made.

The thermograms of the examined child were taken in the supine and backward position from a distance of 1.5 m. In all children, the surface temperatures of selected areas of the body were measured twice (before stimulation and immediately after stimulation with the Vojta method). Before the first registration, the test children were left uncovered for about 10 min in order to equilibrate their body temperature. 

In order to observe changes in the temperature distribution taking place in the subjects, the following system of thermograms was used for qualitative and quantitative analyses: the area of the torso and upper and lower limbs from the front (A1, A2, A3, A4, A5); and from the rear of the torso area and upper and lower limbs (A6, A7, A8, A9, A10) (Figure 1). Via Therma CAM Researcher Pro software 2.10, for each delineated area, the mean surface temperature values were calculated.

### 2.3. Statistical Analysis

The Shapiro–Wilk test was used to check the mean temperature distribution of all analyzed areas, which turned out to be close to normal. Descriptive statistics were calculated. The statistical significance of the mean differences observed between the 1st and 2nd measurements was analyzed with the Student’s *t*-test for dependent groups. The effect of the size of the analyzed relationship was checked by calculating the value of the Cohen’s test d coefficient. The level of significance was *p* < 0.05. The calculations were made using the Statistica 13.3 program and statistical calculators from http://www.psychometrica.de/effect_size.html (accessed on 16 February 2022).

## 3. Results

The performed measurement procedure showed that, after stimulation with the Vojta method, a statistically significant increase in surface temperature was observed on the posterior surface of the children’s body. The temperature values increased in the range 0.57–0.73 °C. The significance of the temperature changes on the posterior surface of the body was confirmed by high values of Cohen’s d coefficient, which indicate a high size effect. The values recorded on the front surface of the body did not change significantly. The recorded difference between the 1st and 2nd measurements was within the range of 0.02–0.22° (Table 1 and Table 2).

## 4. Discussion

Stimulation, according to Vojta, allows for the activation of the muscular corset through the activation of deep muscles. Deep muscles are responsible for keeping the body in an unchanged position while performing activities with the upper and lower limbs. Such activity is necessary to perform gross and fine motor functions as well as orofacial functions [5,7]. Vojta reflex locomotion has been reported to activate the trunk muscles and the deep muscles of the spine to regulate trunk stability and increase spinal rotation force, increasing postural control ability [17,18]. The muscles related to trunk stabilization are generally divided into global muscles and local muscles. The global muscles include the external oblique abdominis, rectus abdominis and erector spinae muscle. The local muscles include the internal oblique abdominis, transversus abdominis and multifidus muscles [19,20,21]. The activities of the local muscles are crucial for increasing trunk stability [22]. During Vojta stimulation, the transversus abdominis muscle is activated first among the abdominal trunk muscles before the extremities move. It generates and regulates abdominal internal pressure together with the contraction of the internal oblique abdominal muscle, the diaphragm and the pelvic floor muscle. It increases the strength of the segmental stability of the lumbar spine to enhance trunk stability [21]. In this study, we observed increasing temperature in the backside as an effect of the activation of muscle chains during Vojta therapy where the stability of the posture is enhanced by the activation of core muscle as a respected reaction. Stabilization through back muscle activation is the target and one of the most important for gross and fine motor development elements.

Physiological motor patterns and the time of their occurrence have been described in the literature [1]. The Vojta method allows for quantitative and qualitative diagnosis and the use of a therapy appropriately selected for the patient’s main problem [23,24,25]. Proper physical activity with the activation of deep muscles affects the circulatory and respiratory systems, improving the efficiency of children and adolescents [26,27,28]. The correct posture of the body predisposes them to undertake physical activity to a large extent as a preventive measure having a long-life effect [18,20].

The change in body temperature after applying an activity is an image of muscle work. In the study group, after the stimulation was applied, a significant temperature change was observed only on the back surface of the body. Activation, through the therapeutic effect of the deep muscles, allowed for the correct activation of the core muscles to program the normative motor pattern. Support extension functions depend on the properties of the muscles that control the body posture in space [12,15,26]. As a non-invasive and reliable method, thermovision can be used for research on various groups, including children and people with disabilities. Similar studies were carried out by Dębiec Bąk, who used thermography to assess body temperature in people with Down syndrome and school-aged children [9,27]. Thermal imaging as a non-invasive method of monitoring the patient’s condition is used in the assessment of neonatal patients in order to monitor adaptation to environmental factors [16]. The increase in skin temperature was also noted in athletes after the completion of bilateral training [14]. Increased skin blood flow leads to increased skin blood volume, due to the arrangement of venous plexuses close to the skin surface. This is directly related to the change in the systolic function of the heart during exercise with the ability of skin blood flow to reach such high levels during body heating necessary to increase convective heat transfer from the body core to the surface of the skin [28]. In the presented study, thermal imaging was used to assess the distribution of body surface temperature based on the emissivity of activated postural muscles through their isometric contraction occurring during Vojta stimulation. The observed change in infrared emissivity occurred as a result of work and muscle contraction. The result of two dynamic physiological processes of heat production and its removal during exercise, changes in blood flow, can be visualized in thermal imaging. Significant differences were shown in the posterior part of the body as a result of the stimulation of postural muscles, especially in the area of the erector spine.

## 5. Conclusions

Thermovision can be a relevant and non-invasive method in the evaluation of muscle work in children aged under 12 months.

## Figures and Tables

**Figure 1 ijerph-19-16072-f001:**
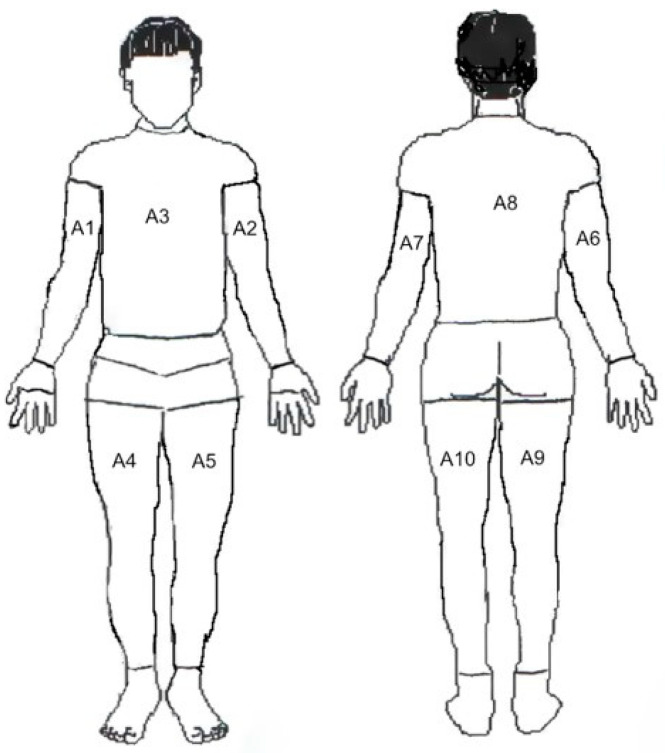
Method of determining the individual fields for thermographic analysis.

**Table 1 ijerph-19-16072-t001:** Anterior surface temperature of the body measured before and after stimulation.

Front of the Body	Research 1		Research 2		Difference	SD Difference	SE Difference	*p* *t* Student’s Test	d Cohen’s Test
	Mean	SD	Mean	SD					
A1 [°]	31.57	0.84	31.56	1.52	0.02	1.29	0.30	0.9570	0.02
A2 [°]	31.68	0.69	31.46	1.45	0.22	1.19	0.27	0.4395	0.27
A3 [°]	33.16	1.12	33.13	1.31	0.03	1.53	0.35	0.9277	0.03
A4 [°]	30.70	1.04	30.64	1.01	0.06	1.41	0.32	0.8565	0.06
A5 [°]	30.71	1.02	30.69	1.03	0.02	1.30	0.30	0.9574	0.02

A1, A2—upper limb, right and left; A3—thorax; A4, A5—lower limb, right and left.

**Table 2 ijerph-19-16072-t002:** Posterior body temperature measured before and after stimulation.

Back of the Body	Research 1		Research 2		Difference	SD Difference	SE Difference	*p* *t* Student’s Test	d Cohen’s Test
	Mean	SD	Mean	SD					
A6 [°]	31.02	0.86	31.76	1.16	−0.73	1.19	0.27	0.0181 *	0.89
A7 [°]	31.02	0.87	31.73	1.19	−0.71	1.05	0.24	0.0103 *	0.98
A8 [°]	32.86	0.75	33.54	1.17	−0.69	1.34	0.31	0.0432 *	0.74
A9 [°]	30.33	1.29	30.89	0.91	−0.57	1.30	0.30	0.0722	0.61
A10 [°]	30.45	1.22	31.09	0.83	−0.65	1.32	0.30	0.0465 *	0.70

A6, A7—upper limb, right and left; A8—thorax; A9, A10—lower limb, right and left; * *p* < 0.05.

## Data Availability

The data presented in this study are available on request from the corresponding author. The data are not publicly available due to [This part of the research are part of a bigger research project].

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
