# Peer review of "Thermographic of the Microcirculation in Healthy Children Aged 3–10 Months as an Objective and Noninvasive Method of Assessment"

_ijerph, 2022, doi:10.3390/ijerph192316072_

Round 1

Reviewer 1 Report (Previous Reviewer 1)

Better, but still there is work done. 

Line 19-20: " ...allows physiotherapists to diagnose local changes based on changes..." - what kind of local changes?

Line 50: "are" sounds too strong. Better  used "maybe"

Line 54: "diastole" refer to cardiac chamber relaxation, not skeletal muscle

Line 62: remove "this"

Line 81 pediatrician and Line 92 Paediatrician - please use one form throughout the text

Discussion

The authors did not respond to earlier stated questions.

"The observed change in infrared emissivity occurred as a result of work, and muscle contraction" - expand: Why were changes only noted on the back side? There is no word for an explanation. Were the exercises used targeted by the back muscles? Could the supine position during exercise possibly cause this observation?

... and in the reviewer's opinion, this is the essence of this work. If the thoracic area was stimulated, how is it that changes in skin temperature were noted only on the back side of the body? Activation of antigravity muscles? muscles of lower limbs? what does the kinematic chain look like during Vojta stimulation? The base of the Vojta method is to release active tension of the muscle chains of both the abdominal and dorsal sides, and then the muscles over the spheroidal joints. Irritation of the brest side among others causes balancing of the lower limbs. Concluding, the discussion should be clearly supplemented with the scheme of activation of muscle chains, because it will give an answer to the question of why the back part of the trunk is characterized by an increase in skin temperature.

Author Response

22/11/2022

Dear Reviewer, thank you for your time and all your supportive remarks. We consider them crucial for the quality of our work. We hope that our explanation which is placed in lines 165-185 will fulfill the gap that we missed in a former version of the manuscript after major revision.

Line 19-20: " ...allows physiotherapists to diagnose local changes based on changes..." - what kind of local changes? The sentence has been changed

Line 50: "are" sounds too strong. Better  used "maybe" – changed for maybe

Line 54: "diastole" refer to cardiac chamber relaxation, not skeletal muscle-removed

Line 62: remove "this"- removed

Line 81 pediatrician and Line 92 Paediatrician - please use one form throughout the text – one form in the whole text

Discussion

The authors did not respond to earlier stated questions. -

"The observed change in infrared emissivity occurred as a result of work, and muscle contraction" - expand: Why were changes only noted on the back side? There is no word for an explanation. Were the exercises used targeted by the back muscles? Could the supine position during exercise possibly cause this observation?

... and in the reviewer's opinion, this is the essence of this work. If the thoracic area was stimulated, how is it that changes in skin temperature were noted only on the back side of the body? Activation of antigravity muscles? muscles of lower limbs? what does the kinematic chain look like during Vojta stimulation? The base of the Vojta method is to release active tension of the muscle chains of both the abdominal and dorsal sides, and then the muscles over the spheroidal joints. Irritation of the brest side among others causes balancing of the lower limbs. Concluding, the discussion should be clearly supplemented with the scheme of activation of muscle chains, because it will give an answer to the question of why the back part of the trunk is characterized by an increase in skin temperature. -

Answer:

Discussion was enriched with explanation of muscle chains activation, which is crucial for explanation of Vojta method - line 165-185

Reviewer 2 Report (Previous Reviewer 2)

Dear Authors:
I have reviewed the manuscript entitled “Thermographic assessment of the microcirculation in healthy children aged 3-10 months subjected to individually selected stimulation elements” for consideration to be published in International Journal of Environmental Research and Public Health.
It is a very interesting study that probably will encourage another authors to perform further studies. Congratulations to the authors.
However, prior to its potential publication, there are some questions that should be reviewed in my opinion.

  • Major aspects:

1. An explanation in the article of the exercises used according to the Vojta method is absolutely necessary in order to clarify that they strongly activate the deep core muscle. The authors were guided by the exercises in the article Epple et al 2020? did they do the same? Did they change something? Please, clarify it!

2. If during the first session the Brest area is mainly used, the authors should be able to tell us in what proportion, was it used in most topics? How many was it used on?

3. The authors must be able to capture this differentiation that can be seen in angular frames, being a therapeutic adaptation to the needs of the patient. Could you indicate how it was captured?

Author Response

22/11/2022

Dear reviewer, thank you for your time  and  giving opinion on aspect that are lack in our work.

Bellow you will find the answer and important notice about our work.

I have re-viewed the manuscript entitled “Thermographic assessment of the microcirculation in healthy children aged 3-10 months subjected to individually selected stimulation elements” for consideration to be published in International Journal of Environmental Research and Public Health.
It is a very interesting study that probably will encourage another authors to perform further studies. Congratulations to the authors.
However, prior to its potential publication, there are some questions that should be reviewed in my opinion.

Dear reviewer, after former revision and suggestion to make major revision of our work we decided to change the title Thermographic of the microcirculation in healthy children aged 3-10 months as an objective and non-invasive method of assessment.  The same we decided that the most important for us in that work is  objective method of observation as an evidence for muscle chains activation during Vojta therapy.

  • Major aspects:
  1. An explanation in the article of the exercises used according to the Vojta method is absolutely necessary in order to clarify that they strongly activate the deep core muscle sentence was removed, the explanation of mechanism is in line 165-185

The authors were guided by the exercises in the article Epple et al 2020? did they do the same? Did they change something? Please, clarify it! – both authors Epple and in our work  used the same methodology of Vojta method recommended by Internationale Vojta Geselshaft. Both study were different in position of leading therapy:  Epple and all worked on side-lying position, authors of this work used supine (back laying)position.  Information were clarify in lines 107-113 z

  1. If during the first session the Brest area is mainly used, the authors should be able to tell us in what proportion, was it used in most topics? How many was it used on? -authors explained it in line 110-112
  2. The authors must be able to capture this differentiation that can be seen in angular frames, being a therapeutic adaptation to the needs of the patient. Could you indicate how it was captured? – after major revision of former version of manuscript we decided not to take under consideration that part and we removed it. We changed title, aim of the study what was seen in changes in main text.

Round 2

Reviewer 2 Report (Previous Reviewer 2)

Dear Authors: I have reviewed the manuscript entitled “Thermographic of the microcirculation in healthy children aged 3-10 months as an objective and non-invasive method of as-sessment” for consideration to be published in International Journal of Environmental Research and Public Health. It is a very interesting study that probably will encourage another authors to perform further studies. Congratulations to the authors

This manuscript is a resubmission of an earlier submission. The following is a list of the peer review reports and author responses from that submission.

Round 1

Reviewer 1 Report

I appreciate the editors and the author the opportunity of reviewing this manuscript.

Abstract: The aim of the study should be clearly stated. The use of the method of thermal imaginary  as the aim per se is of little interest. The research methods are used to investigate, prove, etc.

Introduction:

Lines 57-59 – any confirmation for this statement?

There is no aim of the study pointed out.

I would suggest improving the introduction. Why did the authors use the thermographic imaginary of microcirculation  in Vojta therapy? There is no linking information between the Vojta method and thermal imaging, why might this be useful?

Materials and Methods

The methods should be rewritten. It is not clear whether the camera was calibrated. Do the authors notice that the camera may be calibrated during the test? What was the resolution of the camera? Has the equipment and software used been previously validated in terms of measurement accuracy and its repeatability?

It is not clear whether the study covers a single session or the entire series of meetings. The word therapy suggests that it was a cycle of Vojta sessions, while the description indicates rather a single session.

There is a lack of time for Vojta intervention; it is not known what time after the intervention the thermal measurement was performed; whether it was just a photo or the specified measurement time – all this data should be stated

Discussion:

The first part of the discussion is basically a repetition of an introduction, extended to include later developmental years.

I am not sure whether the stimulation of deep muscles can be assessed on the basis of thermal imaging of the microcirculation of the skin surface – please explain.

Author Response

Answer to review 1

Kindly thank you for your time and all directions. In this way, we could improve our manuscript. Below you will find our answers.

Abstract: The aim of the study should be clearly stated. The use of the method of thermal imaginary as the aim per se is of little interest. The research methods are used to investigate, prove, etc. - Abstracts were improved

Introduction:

Lines 57-59 – any confirmation for this statement? – references were added

There is no aim of the study pointed out. – the aim of the study was  clarify

I would suggest improving the introduction. Why did the authors use the thermographic imaginary of microcirculation in Vojta therapy? There is no linking information between the Vojta method and thermal imaging, why might this be useful? – introduction was improved and enriched in all information

Materials and Methods

The methods should be rewritten. It is not clear whether the camera was calibrated. Do the authors notice that the camera may be calibrated during the test? What was the resolution of the camera? Has the equipment and software used been previously validated in terms of measurement accuracy and repeatability? – the methods part is enriched in all required information

It is not clear whether the study covers a single session or the entire series of meetings. The word therapy suggests that it was a cycle of Vojta sessions, while the description indicates rather a single session.- information was added

There is a lack of time for Vojta intervention; it is not known what time after the intervention the thermal measurement was performed; whether it was just a photo or the specified measurement time – all this data should be stated – information was added

Discussion:

The first part of the discussion is basically a repetition of an introduction, extended to include later developmental years.- the first part was reconstructed, and information was added

I am not sure whether the stimulation of deep muscles can be assessed on the basis of thermal imaging of the microcirculation of the skin surface – please explain. Explanation below:

In the presented study, thermal imaging was used to assess the distribution of body surface temperature based on the emissivity of activated postural muscles through their isometric contraction occurring during Vojta stimulation. The observed change in infrared emissivity occurred as a result of work, and muscle contraction. The resultant of two dynamic physiological processes of heat production and its removal during exercise, changes in blood flow can be visualized in thermal imaging.

Reviewer 2 Report

Dear Authors:
I have reviewed the manuscript entitled “Thermographic assessment of the microcirculation in healthy children aged 3-10 months subjected to individually selected stimulation elements” for consideration to be published in Children
It is a very interesting study that probably will encourage another authors to perform further studies. Congratulations to the authors.
However, prior to its potential publication, there are some questions that should be reviewed in my opinion.

  • Major aspects:

- What type of physiotherapy evaluation was performed?

- The methodologyof the Votja method was selected individually depending on about the main problem diagnosed in the? or, if necesary, make a table about this

Author Response

Answer to review 2

Kindly thank you for your time and remarks. In this way, we could improve our manuscript. Below you will find our answers.

  • Major aspects:

- What type of physiotherapy evaluation was performed? – information is added (line 105-106)

- The methodology of the Vojta method was selected individually depending on the main problem diagnosed in the? or, if necessary, make a table about this – explanation.

All exercises strongly activate the deep core muscle. During the first session, the Brest zone is mainly used. Differentiation is seen in angular settings as a therapeutic adaptation to the patient's needs

Round 2

Reviewer 1 Report

The aim:

Two measurements may show change but not change dynamics.

The observation of something in itself cannot be the goal either. It is still unknown what the reason is for using thermal imaging in Vojta therapy. To prove its effectiveness? As a tool to assess the level of muscle stimulation stimulated with Vojta exercises? Or maybe something else?

Introduction

Please, complete the aim of the study at the end of the introduction. It should be posted not only in the abstract but also at the end of the introduction.

Table 3. Not clear which one side: posterior? or anterior? I guess it is posterior, please correct

Discussion

Needs improvement. The authors should focus more on the main topic of the work.

After rearranging the paragraphs: The first note from the first review now applies to the 3rd paragraph. It is too extensive in relation to the main topic of the article.

Your answer should be posted in the manuscript. 

"The observed change in infrared emissivity occurred as a result of work, and muscle contraction" - expand: Why were changes only noted on the back side? There is no word for an explanation. Were the exercises used targeted at the back muscles? Could the supine position during exercise possibly cause this observation?

Conclusions

are inconsistent with the purpose

Line 315: activation of what?

Author Response

Answer to review 1 round 2

Kindly thank you for your time and all directions.

In this way, we could improve our manuscript.

Sorry if  I misunderstood the former directions.

Below you will find our answers.

The aim:

Two measurements may show change but not change dynamics.- yes, we correct this misleading information

The observation of something in itself cannot be the goal either. It is still unknown what the reason is for using thermal imaging in Vojta therapy. To prove its effectiveness? As a tool to assess the level of muscle stimulation stimulated with Vojta exercises? Or maybe something else?  The aim is correctly written.  Word observation was misleading.

Introduction

Please, complete the aim of the study at the end of the introduction. It should be posted not only in the abstract but also at the end of the introduction. - completed

Table 3. Not clear which one side: posterior? or anterior? I guess it is posterior, please correct - corrected

Discussion

Needs improvement. The authors should focus more on the main topic of the work.

After rearranging the paragraphs: The first note from the first review now applies to the 3rd paragraph. It is too extensive concerning the main topic of the article.- The paragraph was changed a bit, not necessary information was cut out

Your answer should be posted in the manuscript. – the answer is included and expanded.

"The observed change in infrared emissivity occurred as a result of work, and muscle contraction" - expand: Why were changes only noted on the back side? There is no word for an explanation. Were the exercises used targeted by the back muscles? Could the supine position during exercise possibly cause this observation?

Conclusions

are inconsistent with the purpose – are changed to be consistent with the aim

Reviewer 2 Report

I appreciate your answers, just as I would appreciate it if you indicated this explanation in your article. Clarifying what is written is essential so that what they have done can be understood.

1. Reconsider a slight explanation in the article of the exercises used to clarify that they strongly activate the deep core muscle.

2. If during the first session the Brest area is mainly used, the authors should be able to tell us in what proportion, what do they mean in their answer with "mainly used"? Was it used on most subjects? In how many was it used?

3. The authors should be able to capture this differentiation that, as indicated in their response, can be seen in angular frames, being a therapeutic adaptation to the needs of the patient

Author Response

Answer to review 2 round  2

Kindly thank you for your time and remarks. In this way, we could improve our manuscript.

Sorry if  I misunderstood the former directions.

Below you will find our answers.

I appreciate your answers, just as I would appreciate it if you indicated this explanation in your article. Clarifying what is written is essential so that what they have done can be understood – the answer is indicated in the main text to clarify information

  1. Reconsider a slight explanation in the article of the exercises used to clarify that they strongly activate the deep core muscle.- information clryfing are in line 56-60, 205-208
  2. If during the first session the Brest area is mainly used, the authors should be able to tell us in what proportion, what do they mean in their answer with "mainly used"? Was it used on most subjects? In how many was it used?- information was added line 115-119
  3. The authors should be able to capture this differentiation that, as indicated in their response, can be seen in angular frames, being a therapeutic adaptation to the needs of the patient – information was added line 118-119
